# SELF-KNOWLEDGE DISTILLATION ADVERSARIAL ATTACK

## ABSTRACT

Neural networks show great vulnerability under the threat of adversarial examples. By adding small perturbation to a clean image, neural networks with high classification accuracy can be completely fooled. One intriguing property of the adversarial examples is transferability. This property allows adversarial examples to transfer to networks of unknown structure, which is harmful even to the physical world. The current way of generating adversarial examples is mainly divided into optimization based and gradient based methods. Liu et al. (2017) conjecture that gradient based methods can hardly produce transferable targeted adversarial examples in black-box attack. However, in this paper, we use a simple technique to improve the transferability and success rate of targeted attacks with gradient based methods. We prove that gradient based methods can also generate transferable adversarial examples in targeted attacks. Specifically, we use knowledge distillation for gradient based methods, and show that the transferability can be improved by effectively utilizing different classes of information. Unlike the usual applications of knowledge distillation, we did not train a student network to generate adversarial examples. We take advantage of the fact that knowledge distillation can soften the target and obtain higher information, and combine the soft target and hard target of the same network as the loss function. Our method is generally applicable to most gradient based attack methods.

## 1 INTRODUCTION

Neural networks have been shown to be susceptible to adversarial examples. They can be misled when we add small perturbation to a clean image, even though the perturbation may be invisible to the human eye. Adversarial examples are widely used in different physical attack scenarios, including face recognition, voice recognition, and autonomous driving. In the physical world, however, it is difficult to obtain the structure of neural networks. This creates a need for attack algorithms in black-box scenarios. As attack algorithms are created for defense purposes, early discovery of adversarial examples that harm the physical world can help us defend against unknown threats.

Many researchers have observed that adversarial examples can be transferred between different networks. Papernot et al. (2016) create adversarial examples which can be transferred to black-box scenarios by attacking a constructed substitute model. Existing attack algorithms have high attack success rate in white-box scenarios and good transferability in non-targeted attacks. However, there is a very low transferability in targeted black-box scenes which is more harmful. Due to the tricky degree of targeted attack transferability problem, almost all existing researches to improve the transferability focus on the untargeted attack part. In fact, it is easy to understand that an untargeted attack only needs to move the original image away from its category, while a targeted attack needs to reach the target category while penetrating the classification boundary.

Liu et al. (2017) consider gradient based method only searching attacks in a 1-D subspace. In this approach, the subspace contains just a small subset of all target labels. They improve transferability by using ensemble neural networks in the optimization based method. They also prove that the approach of ensemble models is not effective for the gradient based method. For their experimental results and hypotheses, there is no difference between attacking ensemble neural networks and attacking a neural network for the transferability of gradient based method. We will show this is not correct by our study.

There are two problems in existing gradient based methods for targeted attacks. Firstly, the attack success rate of gradient based method cannot be improved after a certain number of iterations. Secondly, targeted attacks do not make full use of the information between different categories.

Existing adversarial attack algorithms usually ensemble neural networks to produce adversarial examples. Dong et al. (2018b) use an approach namely "fuse in logits" to ensemble models. "Fuse in logit" performs a simple weighted average of logits before the softmax layer of the neural network, and then calculates the cross entropy loss between logits and the one-hot vector of the target labels. The ensemble attack algorithm produces adversarial examples in each ensembled model, which can achieve a good attack success rate. But this is not black-box transferability, since every model structure which is ensembled can be acquired by an attack algorithm. Black-box transferability refers to the fact that adversarial examples can still attack successfully against the networks with unknown structures. In this paper, the term transferability will refer uniformly to the black-box and targeted attack scenarios.

Hinton et al. (2015) put forward the concept of knowledge distillation. Knowledge distillation is used to extract useful dark knowledge from bloated models to train a small and lightweight network. Knowledge distillation puts forward the concept of soft targets and hard targets. Through the cumbersome model of probability distribution divided by a value greater than 1, the original probability distribution can be transformed to soft targets. When the soft targets have high entropy, they provide much more information.

Instead of training a small model from multiple large models, we simply take advantage of the fact that knowledge distillation can effectively extract knowledge. Actually, we are not attacking a lightweight model extracted from a bloated model. It is simpler and more efficient to produce transferable adversarial examples in targeted attack, which we call self-distillation adversarial attack.

## 2 BACKGROUND

### 2.1 ADVERSARIAL EXAMPLES

Deep neural networks are proved vulnerable to adversarial attacks since Szegedy et al. (2014), which shows that neural networks can be misclassified by adding small perturbations to a clean picture. The existing of adversarial examples has proven to be a huge threat to the entire application of deep learning algorithms, including speech recognition, face recognition, and even the recognition of road signs in the physical world. As a result, more and more attention has been paid to adversarial examples, and many defensive researches have been made to improve the robustness of neural networks. At present, the research on the attack and defense of neural network is also in a state of game and spiral. The research on adversarial examples is also a good starting points for the explainability of neural network.

### 2.2 GRADIENT-BASED ADVERSARIAL ATTACK METHODS

The current methods of generating adversarial examples are mainly gradient based and optimization based. Here, we briefly introduce the gradient based method.

Goodfellow et al. (2015) come up with a way to get adversarial examples by applying the sign of the gradient to a real example only once. They call this method the Fast Gradient Sign Method (FGSM).

Kurakin et al. (2017) move the adversarial example in the direction of the sign of the gradient in each iteration.

Dong et al. (2018b) integrate momentum into iterative FGSM (I-FGSM), and this can help attack algorithm stabilizing update directions and escaping from poor local maxima. This method which is called MI-FGSM solves the problem that I-FGSM is easy to obtain local extreme value and "overfit" model.

Madry et al. (2018) propose a saddle point formulation, which can convert the attack problems and defense problems to inner maximization problems and outer minimization problems, respectively. They suppose projected gradient descent (PGD) could be the strongest universal attack method.

## 2.3 THE WEAKNESS OF GRADIENT BASED METHOD

The first problem with gradient based methods is that they lose their effectiveness after a certain number of iterations. In neural networks, softmax is usually used as the activation function to get probability output for multi-classification problems. In gradient based adversarial attacks, the probability output of image in a neural network is usually concerned. Therefore, most adversarial attack methods also use softmax activation function to obtain the probability and calculate the cross entropy with the target category, and finally optimize the cross entropy loss function to get the adversarial example.

However, softmax has saturation problems. Softmax's probability output sum is 1, so there is competition among each output. In extreme cases, it becomes winner-take-all, and when one output "approaches 1", the others "approach 0". Another downside of softmax is that it can cause either overspill or underspill problems. Although this downside can be effectively alleviated by subtracting the maximum value of all inputs from the input value of softmax, it causes the output value of softmax to be driven by the maximum input value. When the input value is the maximum, the output value will be close to 1. On the contrary, the large difference can cause softmax's output to be saturated to 0.

Liu et al. (2017) prove that the classification boundaries between white-box models and black-box models do not coincide. And for the model which is not used to generate the adversarial image, its classification area is much smaller. This means that when a successful attack on neural network whose structure is known, adversarial examples cannot penetrates the classification boundaries of other neural networks unknown structure precisely. It is more important to obtain information of different categories. The gradient update stop caused by softmax saturation makes gradient based adversarial examples unable to accurately penetrate classification boundaries.

So, insufficient information acquisition for different categories and premature stop of gradient update are the reasons why gradient based methods have low transferability in targeted black-box scenes. The solution to this problem is very simple. We introduce knowledge distillation into gradient based methods.

## 2.4 DISTILL IN NEURAL NETWORKS

In machine learning tasks, researchers always train many different kinds of models on the same dataset, and combine the results of each single model into a final result (Dietterich (2000)). A simple combined method is to average the predictions, but the cost of this ensemble method is too expensive to deploy.

Hinton et al. (2015) find that instead of simply averaging the predictions, it will get much better performance if converting the logits by:

$$q_i = \frac{exp(z_i/T)}{\sum_j exp(z_j/T)} \tag{1}$$

where $T$ is called a temperature. Using a higher $T$ can produce a softer probability distribution over classes. And it will be normal softmax-layer when $T$ is set by 1.

Distillation in neural networks can make the combination of multiple networks perform better, and the knowledge of the cumbersome model can tansfer to a small model in an effective way. The current research on distillation is mainly used to train a smaller model from a bloated model set. However, distillation can effectively help to get the target information in adversarial attacks.

## 3 METHODOLOGY

### 3.1 THE SOLVE OF SATURATION PROBLEM

Here, we prove that knowledge distillation can effectively reduce the distance between classes and solve the saturation problem. If we use the ratio $\rho$ to measure the distance between the softmax-layer's outputs, we can get the distance of two hard targets below.

$$\rho = \frac{q(x_a)}{q(x_b)} = \frac{\frac{e^{x_a}}{\sum_i e^i}}{\frac{e^{x_b}}{\sum_i e^i}} = \frac{e^{x_a}}{e^{x_b}} \tag{2}$$

If we distill the logits with a temperature $T$ like formula-1, the distance will be as follows.

$$\rho' = \frac{q(x_a/T)}{q(x_b/T)} = \frac{\frac{e^{x_a/T}}{\sum_i e^i/T}}{\frac{e^{x_b/T}}{\sum_i e^i/T}} = \frac{e^{x_a/T}}{e^{x_b/T}} \tag{3}$$

So the relationship between $\rho'$ and $\rho$ is:

$$\rho' = \sqrt[T]{\rho} \tag{4}$$

The distillation processing is essentially the dispersion of the softmax results among different classifications using the $T$-th power of the square root, so that the different classification softmax values distribution are closer.

## 3.2 GET MORE ACCURATE DIRECTION

It may be a good thing for untargeted attacks when data classes are close to each other, because it makes easy for adversarial examples to randomly move from the original category to a target category and the attack succeeds. But for targeted attacks, this is a problem, making it more likely to attack the wrong class.

Liu et al. (2017) observe that the decision boundaries of all models are very consistent with each other. They think this explains why non-targeted adversarial images can transfer among models. But for targeted attack, decision boundaries of models do not generate adversarial examples in a small area, thus tiny change of gradient will make adversarial examples change to wrong target categories. We assume that a gradient based method can obtain a more accurate gradient direction by knowledge distillation.

The distance between the gradient of a target class and the gradient of a wrong class can be give by:

$$\Delta = \frac{\partial C_{target}}{\partial z_i} - \frac{\partial C_{wrong}}{\partial z_i} = (p_{True} - q_{target}) - (p_{True} - q_{wrong}) = q_{wrong} - q_{target} \tag{5}$$

where $C_{target}$ means the cross entropy about target class, $C_{wrong}$ means the cross entropy about wrong probability. $\Delta$ means the distance of two gradient. We define $\delta = \Delta/p_{true}$, so it is easy to get:

$$
\begin{aligned}
\phi &= \frac{\delta'}{\delta} = \frac{\Delta'/p'_{true}}{\Delta/p_{true}} = \frac{(q'_{wrong} - q'_{target})/p'_{true}}{(q_{wrong} - q_{target})/p_{true}} \\
&= \frac{\frac{e^{wrong/T} - e^{target/T}}{\sum_i e^i/T}/\frac{e^{true/T}}{\sum_i e^i/T}}{\frac{e^{wrong} - e^{target}}{\sum_i e^i}/\frac{e^{true}}{\sum_i e^i}} = \frac{e^{wrong/T} - e^{target/T}}{e^{true/T}}/\frac{e^{wrong} - e^{target}}{e^{true}}
\end{aligned} \tag{6}
$$

Compared to the true class, the values of the wrong class and the target class tend to zero. We can approximate that:

$$\phi \approx \frac{wrong/T - target/T}{e^{true/T}}/\frac{wrong - target}{e^{true}} \approx \frac{e^{true}}{e^{true/T}} >> 1 \tag{7}$$

It means distillation can effectively increase the distance between the gradient of the target class and the gradient of a wrong class. In this way, the gradient based attack method can attack to a targeted class more precisely.

# 4 DISTILLATION ADVERSARIAL ATTACK

## 4.1 SINGLE MODEL GRADIENT-BASED ATTACK

Any gradient based adversarial attack algorithm can be easily extended to include distillation. For example, FGSM can be simply extended to Distillation FGSM (D-FGSM) as follows.

$$X^{adv} = x^{real} + \epsilon \cdot sign(\bigtriangledown_x J(\frac{L(x)}{T}, y))$$

Here, $\bigtriangledown_x J$ is the gradient of loss function, L(x) means the logits of image x produced by neural network, and $sign(\cdot)$ is sign function limits the size of the disturbance.

Momentum iteration fast gradient sign method can be simply extended to Distillation MI-FGSM (D-MIFGSM).

$$g_{t+1} = \mu \cdot g_t + \frac{\nabla_x J(\frac{L(x)}{T}, y)}{\| \nabla_x J(\frac{L(x)}{T}, y) \|_1} \qquad x_{t+1}^{adv} = x_t^{adv} + \alpha \cdot sign(g_{t+1})$$

## 4.2 ENSEMBLE BASED METHOD

Liu et al. (2017) ensemble neural networks to improve transferability in optimize based method. They consider there is no difference between single model attack and ensemble based attack in fast gradient based method, and both these two methods can not produce transferable adversarial examples. Dong et al. (2018b)'s ensemble method is a simple weighted average of logits generated by multiple neural networks, which they call "fuse in logits".

We have a better ensemble approach. By distilling logits generated by $n$ neural networks, $n$ softened logits can be obtained. In the gradient based attack method, $n$ logits can obtain $n$ cross entropy losses, and the cross entropy losses generated by the softened logits and the original logits, namely the hardened logits, can produce a total of $2n$ cross entropy losses. A simple weighted average of these cross entropy losses can extend the gradient based approach to the ensemble attack method that includes knowledge distillation:

$$J(L(x), y) = \lambda_1 \cdot \sum_{i=1}^{n} J(\frac{L(x_i)}{T}, y) + \lambda_2 \cdot \sum_{i=1}^{n} J(L(x_i), y) \tag{8}$$

where $L(x)$ is the logit produced by different neural networks, and $J(x, y)$ is the cross entropy between the logit $x$ and the attack target $y$.

We have carried out experiments on different adversarial attack algorithms and found that the momentum iterative fast gradient sign method based on knowledge distillation (D-MIFGSM) is the most effective method to produce transferable adversarial examples.

We summarize our ensemble based method in Algorithm 1, and the rest of our experiments are based on this algorithm. Specifically, when we input a clean image $x$ and get $K$ logits from $K$ classifiers, we distill these $N$ logits using a pre-defined temperature $T$. Then, we perform Equation 8 "fuse in cross entropy" on these distilled logits. The rest of the algorithm is the same as MI-FGSM.

# 5 EXPERIMENT

## 5.1 SETUP

To see how distillation works on targeted attack and what is the best distillation environment, we design a series of experiments. We choose four networks, $Inception\_V3$, $Inception\_V4$ (Szegedy

---

**Algorithm 1** Ensemble D-MIFGSM

---

Input: The logits of $K$ classifiers $l_1, l_2, ..., l_K$; Temprature $T$; a real example $x$ and target label $y^*$;
Input: The size of perturbation $\varepsilon$; iterations $N$ and decay factor $\mu$.
Output: An adversarial example $x^*$ with $\parallel x^* - x \parallel_{\inf} \leq \varepsilon$
1: $\alpha = \varepsilon / N$ ;
2: for $i = 0$ to $N - 1$ do
3: Input $x_i^*$ and output $l_k(x_i^*)$ for $k = 1, 2, ..., K$;
4: Get different logits as $l(x_i^*) = \begin{cases} \frac{l(x_i^*)}{T} \\ l(x_i^*) \end{cases}$
5: Fuse the softmax cross-entropy loss $J(x_i^*, y^*)$ based on Equation 8.
6: Obtain the gradient $\nabla_x J(x_i^*, y^*)$;
7: Update $g_{i+1}$ by accumulating the velocity vector in the gradient direction as $g_{i+1} = \mu \cdot g_i + \frac{\nabla_x J(x_i^*, y^*)}{\parallel \nabla_x J(x_i^*, y^*) \parallel_1}$;
8: Update $x_{i+1}^*$ by applying the sign gradient as $x_{i+1}^* = x_i^* - \alpha \cdot sign(g_{i+1})$;
9: end for
10: return $x^* = x_N^*$ .

---

et al. (2016)), $VGG16$ and $VGG19$ (Simonyan & Zisserman (2015)), to produce the adversarial examples. We use $ResNet\_V1\_50$, $ResNet\_V1\_152$ (He et al. (2016)) and $Inception\_Resnet$ (Szegedy et al. (2017)) as the networks in black-box attacks to test the transferability of the adversarial examples.

To verify the true effectiveness, we tend to choose a bigger dataset. So, we randomly choose 10,000 images from ILSVRC 2012 validation set, with 10 images in each class, and make sure they can be all classified correctly by the networks we use. We define the disturb of adversarial examples by $L_{\inf}$-distance.

## 5.2 SINGLE MODEL ATTACK

For Liu et al. (2017), attacking one model is the same as attacking multiple models in fast gradient method, neither of which can produce adversarial examples with high transferability. We test the performance of different networks in a single model attack. Table 1 shows that the transferability in single model attack can reach about $30\%$ by attacking $VGG$ networks. And we find that only attacking $VGG$ networks can produce transferable adversarial examples.

|         | $IncV3$ | $IncV4$ | $VGG16$ | $VGG19$ | $Resnet\_50$ | $Resnet\_152$ | $Inc\_Res$ |
|---------|---------|---------|---------|---------|--------------|---------------|------------|
| $IncV3$ | 0.999   | 0.018   | 0.009   | 0.007   | 0.006        | 0.005         | 0.020      |
| $IncV4$ | 0.059   | 0.991   | 0.125   | 0.010   | 0.005        | 0.005         | 0.027      |
| $VGG16$ | 0.241   | 0.205   | 1.0     | 0.929   | 0.340        | 0.235         | 0.201      |
| $VGG19$ | 0.233   | 0.197   | 0.918   | 1.0     | 0.333        | 0.241         | 0.191      |

Table 1: The success rate of D-MIFGSM in single model attack. The cell $(i, j)$ indicates the accuracy of the adversarial examples generated for model $i$ (row) evaluated over model $j$ (column).

## 5.3 ENSEMBLE BASED METHOD

We find that attacking $VGG$ networks can generate adversarial examples with high transferability in a single model attack. Moreover, in the optimization-based approach, ensemble networks have been proved to be able to produce transferable adversarial examples.

We test the impact of ensemble on transferability of different networks. In our test, we take off one of the four known networks each time, and the results are shown on Table 2.

Since it is possible to generate a large number of adversarial examples with high transferability in the distillation based method through ensemble models, we perform different parameter tests on this method.

## 5.4 THE EFFECT OF TEMPERATURE

Firstly, we want to know how different distillation temperatures affect the attack success rate, and which is the best temperature to distill. We test different temperatures while the iteration number is 20 and the max disturb limit is 32.

|       | $IncV3$ | $IncV4$ | $VGG16$ | $VGG19$ | $Resnet\_50$ | $Resnet\_152$ | $Inc\_Res$ |
|-------|---------|---------|---------|---------|--------------|---------------|------------|
| $All$ | 0.949 | 0.982 | 1.0 | 1.0 | 0.471 | 0.385 | 0.334 |
| $-IncV3$ | 0.499 | 0.991 | 1.0 | 1.0 | 0.575 | 0.462 | 0.386 |
| $-IncV4$ | 0.937 | 0.294 | 1.0 | 1.0 | 0.437 | 0.355 | 0.219 |
| $-VGG16$ | 0.939 | 0.980 | 0.803 | 1.0 | 0.232 | 0.176 | 0.192 |
| $-VGG19$ | 0.941 | 0.979 | 1.0 | 0.787 | 0.228 | 0.173 | 0.188 |

Table 2: The success rate of D-MIFGSM in ensemble based method. Cell $(i, j)$ indicates that the accuracy of the targeted adversarial examples generated for the ensemble of the four models except model $i$ (row) is predicted as the target label by model $j$ (column). In each line , $-$ means the network behind is not used in ensemble attack.

Figure 1 shows the effect of different distillation temperatures on black-box targeted attack, where the temperature equals 1 means there's no distillation processing. It is clear that the distillation can make effects on both single model attack and ensemble based method. And the distillation temperature makes best effects when it reaches about 16.

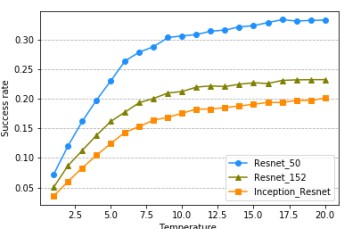 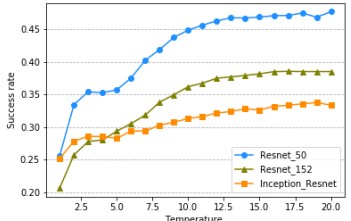

(a) Different Temperature in Single Models

(b) Different Temperature in Ensemble Models

Figure 1: Effect of Different Temperature

## 5.5 THE EFFECT OF THE ITERATION

After the test of temperature, we want to know the effect of the iteration number on the success rate of the attack in the distillation method. Dong et al. (2018a) suppose that the adversarial examples generated by iterative methods have a tendency to overfit, which means iteration is negative to the transferability of adversarial examples.

But in our method, as Figure 2 shows, the more iterations, the much transferability we can get, until it is large enough to converge. In another word, to certain extent, we avoid the overfit problem.

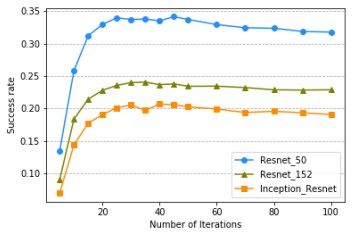 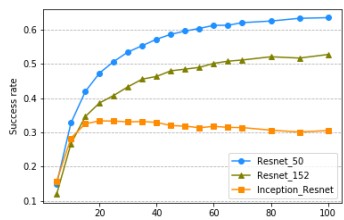

(a) Different Iterations in Single Models

(b) Different Iterations in Ensemble Models

Figure 2: Effect of Different Iterations

However, high iterations also mean long computing time. In our experiment, we suggest 30 to 50 iterations to be a better choice.

## 5.6 THE EFFECT OF THE DISTURB LIMIT

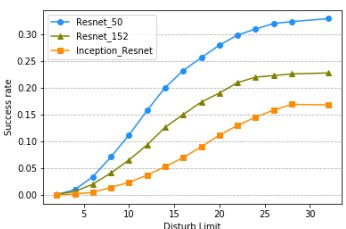 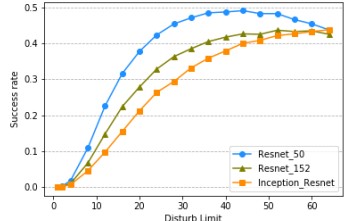

(a) Different Disturb Limit in Single Models

(b) Different Disturb Limit in Single Models

Figure 3: Effect of Different Disturb Limit

The distance between origin images and adversarial examples is also an important indicator to judge the performance of attacking methods. In both MIFGSM and D-MIFGSM, the distance depends on the max disturb limit which is a hyperparameter in attack processing. It is also an important task to find out the best disturb limit.

As Figure 3 shows, lower disturb causes lower attack success rate. But at the mean time, too big disturb limit also has a negative effect on the success rate of adversarial attack. On the other hand, bigger disturb means bigger noise, which is more likely to be realized by human. Above all, in our experiment, we prefer to choose the disturb limit between 16 and 32.

## 5.7 THE COMPARISON BETWEEN MIFGSM AND D-MIFGSM

We compare the attack success rate in both black-box and white-box attack scenarios between MIFGSM and our D-MIFGSM. The setup of the temperature is 16, the iterations is 20 and the disturb limit is 32.

Table 3 shows that in both two kinds of attack scenarios D-MIFGSM provides a significant boost to MI-FGSM.

| | | Black-Box | | | White-Box | | | |
|---|---|---|---|---|---|---|---|---|
| | | $Resnet\_50$ | $Resnet\_152$ | $Inc\_Res$ | $IncV3$ | $IncV4$ | $VGG16$ | $VGG19$ |
| Single | $MIFGSM$ | 0.072 | 0.050 | 0.002 | — | — | 1.0 | — |
| Model | $D-MIFGSM$ | 0.333 | 0.231 | 0.191 | — | — | 1.0 | — |
| Ensemble | $MIFGSM$ | 0.201 | 0.154 | 0.170 | 0.944 | 0.789 | 0.993 | 0.988 |
| Models | $D-MIFGSM$ | 0.471 | 0.385 | 0.334 | 0.982 | 1.0 | 1.0 | 0.949 |

Table 3: The success rate of MIFGSM and our D-MIFGSM. The single model attack use $VGG16$ to generate adverarial examples. Ensemble based method use all four models to generate adverarial examples in 5.1.

## 6 CONCLUSION

In this paper, we show that gradient based methods can also produce adversarial examples with high transferability. In order to generate adversarial examples with high transferability in targeted attack, we introduce knowledge distillation into gradient based methods. Our method can be easily implemented in any method based on fast gradient. In targeted attack, the ensemble of multiple networks and optimization based methods is proved to produce adversarial examples with high transferability. But optimization-based methods are notoriously slow. We introduce knowledge distillation into the momentum iterative fast gradient method to produce adversarial examples of about 30% to 50% transferability in the experiment. At the same time, we find out that the network structure also has an effect on transferability, and it will be our future work to explore neural network structures that can produce transferability.

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
