# OpenReview forum: "SELF-KNOWLEDGE DISTILLATION ADVERSARIAL ATTACK"
_ICLR.cc/2020/Conference — Reject_

### Official Review · AnonReviewer3 · 2019-10-21
**Official Blind Review #3**

**Rating:** 1

**Review:**

This paper proposes an attack method to improve the transferability of targeted adversarial examples. The proposed method uses a temperature T to convert the logit of the network, and calculates the gradient based on the new logit, yielding the distillation-based attack method. It has been integrated into FGSM and MI-FGSM.

Overall, this paper has the poor quality based on the writing, presentation, significance of the algorithm, insufficient experiments. The detailed comments are provided below.

1. The writing of this paper is poor. There are a lot of typos in the paper. The notations are used without definitions. These make the paper hard to read and understand.

2. Based on my understanding of the paper, the motivation of the proposed method is that the softmax function on top of neural networks can make the gradient unable to accurately penetrate classification boundaries. And the distillation-based method is proposed to reduce the magnitude of the logits to make the gradient more stable. However, if the argument were true, we could use the C&W loss to perform the attack, which is defined on the logit layer without affected by the softmax function.

3. There are a lot of recent attack methods proposed to improve the transferability of adversarial example, e.g., "Improving transferability of adversarial examples with input diversity" (Xie et al., 2019); "Evading defenses to transferable adversarial example by translation-invariant attacks" (Dong et al., 2019). The authors are encouraged to compare the proposed methods with previous works.

**Experience Assessment:**

I have published in this field for several years.

**Review Assessment: Checking Correctness Of Derivations And Theory:**

I carefully checked the derivations and theory.

**Review Assessment: Checking Correctness Of Experiments:**

I carefully checked the experiments.

**Review Assessment: Thoroughness In Paper Reading:**

I read the paper thoroughly.

---

> ### Author Response · Authors · 2019-11-14
> **Thanks for your review**
>
> Q: It seems the softmax layer can be avoided, like CW attack.
> A: Thank you for this suggestion. Compare to gradient-based method, CW can get better attack success rate, but it takes too much time to attack, and the L-2 distance and L-inf distance is far bigger. What we do is to increase the transferability of adversarial examples without the increase of attack time and distance, so CW is not a good choice to our work.
>
> Q: There are a lot of recent attack methods proposed to improve the transferability of adversarial example, like (Xie et al., 2019) and (Dong et al., 2019).
> A: There are some works on the improvement of the transferability of adversarial examples before, but these works focus on the untargeted attack. In fact, the transferability in the targeted attack, especially gradient-base attack, is known as impossible before. What we do is the breakthrough of the transferability of targeted attack.

---

### Official Review · AnonReviewer2 · 2019-10-21
**Official Blind Review #2**

**Rating:** 3

**Review:**

This paper proposes distillation attacks to generate transferable targeted adversarial examples. The technique itself is pretty simple: instead of only using the raw logits L(x) to compute the cross entropy loss for optimization, they also use the distilled logits L(x)/T to generate adversarial examples. Their evaluation setup largely follows the style of Liu et al., but they construct a different subset of ILSVRC validation set, and some of the model architectures in their ensemble are different from Liu et al. Their results show that by including the distilled logits when computing the gradient, the generated adversarial examples can transfer better among different models using both single-model and ensemble-based attacks.

I think their proposed attack is interesting due to its simplicity and effectiveness. However, I would like to see clarification of some evaluation details, as well as more experiments to compare with Liu et al.:

1. To assess the effectiveness of targeted attacks, it is important to ensure that the semantic meaning of target label is far from the ground truth label. Some of the 1000 ImageNet labels have very similar meanings to each other, thus different choices of the target label would dramatically affect the difficulty of the attacks. In Liu et al., they manually inspect the image-target pairs to ensure that the target label is very different from the ground truth in its meaning. To enable a fair comparison, it would be helpful to provide results on the same image-target pairs constructed by Liu et al., which could be found in the public repo linked in their paper.

2. For ensemble attacks, is including both the raw and the distilled logits crucial in obtaining a good performance? What is the performance of including distilled logits only? How do different values of \lambda_1 and \lambda_2 in (8) affect the attack performance?

3. Could you visualize some generated adversarial examples, so that we can view the qualitative results?

4. In general this paper lacks empirical analysis on why distillation helps improve the transferability. Some more discussion would be helpful.

-------------
Post-rebuttal comments

Thanks for your response! I think this paper still misses a more in-depth analysis, and thus I keep my original assessment.
-------------

**Experience Assessment:**

I have published in this field for several years.

**Review Assessment: Checking Correctness Of Derivations And Theory:**

I carefully checked the derivations and theory.

**Review Assessment: Checking Correctness Of Experiments:**

I carefully checked the experiments.

**Review Assessment: Thoroughness In Paper Reading:**

I read the paper thoroughly.

---

> ### Author Response · Authors · 2019-11-14
> **Thanks for your review**
>
> Q: The target labels should be far from the true labels.
> A: Thanks! It's really a great suggestion. In our experiments, we all randomly generate the true labels. Statistically speaking, the probability of a similar labels is very small. Although this effect cannot be completely ruled out, the results should be credible. We also manually check our experimental data to ensure that the impact of such problems is minimized.
>
> Q: Is including both the raw and the distilled logits crucial in obtaining a good performance? What is the performance of including distilled logits only? How do \lambda_1 and \lambda_2 in (8) affect the attack performance?
> A: Another great question! In our experiments, only include the distilled logits to generate the cross entropy is not a good idea. Since the distillation makes the distance of true labels and target labels, and it makes the attack will stop when the attack distance is enough to the new distance, but not enough to the origin distance. The lack of attack distance also cause the lower transferability. So we include both the raw and the distilled logits to generate the cross entropy, to make up the attack distance.
> Actually, the different value of \lambda_1 and \lambda_2 in (8) do affect, but the affect is not as significant as we expected, and we are looking for the reason.

---

### Official Review · AnonReviewer1 · 2019-10-28
**Official Blind Review #1**

**Rating:** 3

**Review:**

The paper suggests to use temperature scaling in adversarial attack design for improving transferability under black-box attack setting. Based on this, the paper proposes several new attacks: D-FGSM, D-MIFGSM, and their ensemble versions. Experimental results found that the proposed methods improves transferability from VGG networks, compared to the non-distillated counterparts.

In overall, I liked its novel motivation and simplicity of the method, but it seems to me the manuscript should be improved to meet the ICLR standard. Firstly, the presentation of the method is not that clear to me. The mathematical notations are quite confusing for me as most of them are used without any definitions. I am still not convinced that the arguments in Section 3.1 and 3.2 are indeed relevant to the actual practice of black-box adversarial attacks, which usually includes extremely non-smooth boundaries with multiple gradient steps. Even though the experiments show effectiveness partially on VGGNets, but the overall improvements are not sufficient for me to claim the general effectiveness of the method unless the paper could provide additional results on broader range of architectures and  threat models.

- I feel Section 2.3 is too subjective with vague statements. The following statement was particularly unclear to me: "The first problem with gradient based methods is that they lose their effectiveness after a certain number of iterations.": Does the term "effectiveness" indicate some relative effectiveness compared to other methods, e.g. optimization-based attacks? Is this really a general phenomenon in gradient-based attacks? Also, please elaborate more on "So, insufficient information acquisition for different categories and premature stop of gradient update are the reasons ..."

- Regarding that the softmax is the problem, one could try to directly minimize the logit layers skipping the softmax, i.e., gradient on logits? This is actually one of common techniques and there are many simple tricks in the context of adversarial attack, so the paper may include comparisons with such of tricks as well.

- It is important to specify the exact threat model used throughout the experiments, e.g. perturbation constraints and attack details. Demonstrating the effectiveness on a variety of threat models could also strengthen the manuscript.

- Table 1 and 2 may include other baseline (black-box attack) methods for comparison. This would much help to understand the method better.

**Experience Assessment:**

I have read many papers in this area.

**Review Assessment: Checking Correctness Of Derivations And Theory:**

I assessed the sensibility of the derivations and theory.

**Review Assessment: Checking Correctness Of Experiments:**

I assessed the sensibility of the experiments.

**Review Assessment: Thoroughness In Paper Reading:**

I read the paper at least twice and used my best judgement in assessing the paper.

---

> ### Author Response · Authors · 2019-11-14
> **Thanks for your review**
>
> Thanks a lot. We think you did read our manuscript very carefully. Here are our responses to your major concerns.
> Q: The first problem with gradient based methods is that they lose their effectiveness after a certain number of iterations.": Does the term "effectiveness" indicate some relative effectiveness compared to other methods, e.g.
> A: In fact, “effectiveness” is compared with any non-distilled version of the gradient-based method. In our experiment, we found that our method could make the gradient-based method iterate more times and bring about improvement, which is really a pity that we did not put this part of the experimental results in our manuscript.
>
> Q:It is important to specify the exact threat model used throughout the experiments, e.g. perturbation constraints and attack details. Demonstrating the effectiveness on a variety of threat models could also strengthen the manuscript.
> A: In Section 5.7, we have written the experimental Settings such as noise level, temperature, etc. However, we did ignore the very important setting instructions in some places. In fact, in the experiments we compared with other methods, the noise was 32 and the temperature was about 16.
> Thank you very much for your pertinent suggestions!

---

### Author Response · Authors · 2019-10-21
**The code of this paper**

Sorry that the code is not be uploaded with the paper, and here is the code link (share by Google Drive):

https://drive.google.com/open?id=10bSk9u3iBbSu_gnG0UD6HE56qQA2-R-r

---

### Public Comment · ~Anthony_Wittmer1 · 2019-10-21
**About the motivation**

Hi, it is an interesting work.

I have some confusion about the motivation, i.e., why increasing the distance between the gradient of the target class and the gradient of a wrong class can help the adversary attack more precisely? What does the distance between the gradients mean?

In addition, the expression L(x)/T has sonething similar with the logits fusing w_k*L_k(x) in [1], where w_k is in the range of [0, 1].

[1] Boosting Adversarial Attacks with Momentum, CVPR 2018

---

> ### Author Response · Authors · 2019-10-22
> **About the Questions**
>
> Thanks.
> Firstly, about the increasing distance. What we do is not to only increase the distance. The distillation we make is to lower the saturation, make the difference much bigger, and confirm the targeted attack into target class(For all models). (In paper Section 3)
> And the other question, about the $w_k$ in MI-FGSM. In MI-FGSM, the multiple models' logits are fused in one logits, the $w_k$ is the weights of each model, and $\sum{w_k}=1$. In this work, we do not fuse the logits. T is the temperature in distiallation, to control how much lower the saturation we want.  It's totally different with MI-FGSM's work.

---

### Public Comment · ~Michael_Matthew1 · 2019-10-22
**Question about the defense models**

This is an interesting work. Though, I like the idea for your approach, I have a few concerns. It seems that only the target attack on the clean model is considered in the paper.  How would the results look like when attacking the defense models (e.g. adversarial training models[1], JPEG[2] and TVM[2]).

References:
[1] Ensemble adversarial training: Attacks and defenses. ICLR 2018.
[2] Countering adversarial images using input transformations. ICLR 2018.

---

> ### Author Response · Authors · 2019-10-23
> **About the defense models**
>
> Thank you for your comments!
> In fact, this works's idea is inspired by an attack and defensive confrontation competition. We use this method to attack others' defense models and get a good results. Even face to those defense models, we also can get transferable adverserial examples in black-box targeted attack.
> So firstly, we work on why it can get transferable adverserial examples in black-box targeted attack, and that is what this paper talking about. After all, it is used to be conjectured that gradient based methods can hardly produce in black-box attack[1].
> In fact, our method can be combined with attack defense model methods such as TI-MIFGSM[2] and DIM[3].
>
> References:
> [1]Delving into transferable adversarial examples and black-box attacks.  ICLR 2017
> [2]Evading Defenses to Transferable Adversarial Examples by Translation-Invariant Attacks. CVPR2019
> [3]Improving Transferability of Adversarial Examples With Input Diversity. CVPR2019

---

### Public Comment · ~Micah_Goldblum1 · 2019-11-08
**An Interesting Connection**

Hi Authors,
Thank you for your interesting paper.  I noticed that your work concerning adversarial attacks via distillation is related to our work on producing adversarially robust networks using distillation.[1]  Please consider mentioning the relationship with our work in your next version.

[1] Goldblum, Micah, et al. "Adversarially Robust Distillation." arXiv preprint arXiv:1905.09747 (2019).

---

> ### Author Response · Authors · 2019-11-14
> **About the similarities**
>
> Thank you very much for your comments. We have read your paper briefly, and it is true that there are similarities as you say, and further support the idea of training a distillation network to produce adversarial examples as mentioned in our future work. We will carefully compare your work in our future work.

---

### Decision · Program_Chairs · 2019-12-19

**Decision:**

Reject

**Comment:**

This paper proposes an attack method to improve the transferability of adversarial examples under black-box attack settings.

Despite the simplicity of the proposed idea, reviewers and AC commonly think that the paper is far from being ready to publish in various aspects: (a) the presentation/writing quality, (b) in-depth analysis and (c) experimental results.

Hence, I recommend rejection.